# Numerical and Experimental Results on Charpy Tests for Blends Polypropylene + Polyamide + Ethylene Propylene Diene Monomer (PP + PA + EPDM)

**DOI:** 10.3390/ma13245837

**Published:** 2020-12-21

**Authors:** Cătălin Pîrvu, Andreea Elena Musteată, George Ghiocel Ojoc, Lorena Deleanu

**Affiliations:** 1National Institute of Aerospace Research “Elie Carafoli” INCAS, 061126 Bucharest, Romania; 2Faculty of Engineering, “Dunărea de Jos” University of Galati, 800008 Galati, Romania; andreea.musteata@ugal.ro (A.E.M.); george.ojoc@ugal.ro (G.G.O.)

**Keywords:** Charpy test, PA6, PP, EPDM, simulation, FE analysis, constitutive model, tensile test

## Abstract

This paper presents results from numerical and experimental investigation on Charpy tests in order to point out failure mechanisms and to evaluate new polymeric blends PP + PA6 + EPDM. Charpy tests were done for initial velocity of the impactor of 0.96 m/s and its mass of 3.219 kg and these data were also introduced in the finite element model. The proposed model takes into account the system of four balls, including support and the ring of fixing the three balls and it has a finer discretization of the impact area to highlight the mechanisms of failure and their development in time. The constitutive models for four materials (polypropylene with 1% Kritilen, two blends PP + PA6 + EPDM and a blend PA6 + EPDM) were derived from tensile tests. Running simulations for each constitutive model of material makes possible to differentiate the destruction mechanisms according to the material introduced in the simulation, including the initiation and the development of the crack(s), based on equivalent plastic strain at break (EPS) for each material. The validation of the model and the simulation results were done qualitatively, analyzing the shape of broken surfaces and comparing them to SEM images and quantitatively by comparing the impact duration, energy absorbed by the sample, the value of maximum force during impact. The duration of the destruction of the specimen is longer than the actual one, explainable by the fact that the material model does not take into account the influence of the material deformation speed in Charpy test, the model being designed with the help of tests done at 0.016 m/s (1000 mm/min) (maximum strain rate for the tensile tests). Experimental results are encouraging for recommending the blends 20% PP + 42% PA6 + 28% EPDM and 60% PA6 + 40% EPDM as materials for impact protection at low velocity (1 m/s). Simulation results are closer to the experimental ones for the more brittle tested materials (with less content of PA6 and EPDM) and more distanced for the more ductile materials (with higher content of PA6 and EPDM).

## 1. Introduction

Charpy tests have been characterized materials for impact resistance till the XIX century [1,2,3,4,5], but relevance had been initiated by Charpy [6] the test keeping his name till nowadays. 

Reviewing the available literature on this type of tests [7,8,9,10] and on Charpy impact modeling [11,12,13,14,15,16,17] pointed out the following aspects:-models are using very different mesh web rougher or finer (especially near the notch) [18,19,20], on a quarter or half of the sample, from 2D to 3D [21], but new achievement in hardware and dedicated software allows for using finer mesh, all the sample, investigation on strain and stress, simulation in actual time etc.;-first models did not take into account friction, but recent ones have included condition for friction, usually with a constant friction coefficient;-at the beginning, constitutive models were simpler, implying only a failure criterion, usually the strength limit in tensile and there has been interest in performant materials like steel, titanium and aluminum alloys [22,23]; for polymers and especially for blends fewer research reports being [24,25,26,27]; and-each model has simplifying hypotheses that could alter the sample behavior as compared to the actual one.

Many engineers were interested in Charpy test for metallic materials [11,28,29,30], but nowadays this test, in classical or modified configuration, becomes useful to be done for other materials, including polymers, composites [31], and ceramics [32]. Simulation of such a test is useful for understanding the material behavior under impact load, and when knowing mechanical characteristics at impact, to assess the first step in designing structures with tested materials under such conditions.

Tanguy [11] proposed a simulation with a detailed description of the material viscoplastic behavior over a wide temperature range. The Charpy test is simulated using a full 3D mesh and accounting for adiabatic heating and contact between the specimen, the striker and the anvil. This research team stipulated that 2D plane strain simulations instead of 3D ones offer a significant reduction of running time, but an overestimation of load and, consequently, the crack advance. The model is well suited to represent ductile tearing. Among material factors affecting the material model to be introduced for simulating Charpy test are: The simplifying concept for the model (2D or 3D, recent works being in the favor of the last, including the simulation here presented), accepting quarter or half of the actual test system (imposing symmetrical planes that are not supported by the results of this work), conditions for constraints among parts of the systems (no contact with anvils can be done by imposing a fixed displacement on the opposite side of the notch and a zero displacement on the initial contact line between the anvil and the specimen, but this approach develops higher stresses in the notch, causing earlier ductile crack start), adiabatic heating and heat diffusion, the viscoplastic behavior of the material (at higher impact velocities, ultimate strength is increased, which causes earlier failure). 

Smith et al. [33] did a FE (finite element) analysis of Charpy specimen (both quasi-static and dynamic loading), using a symmetrical quarter model and an original mesh. The notch was modeled with a radius of 0.25 mm and internal angle of 45° and, at the tip of the notch, 30 concentric rings of elements were defined. The model was validated by comparing the load vs. clip gauge opening values from the simulation to experimental values. The plastic behavior of this material was modeled using von Mises plasticity with isotropic hardening. The dynamic loading condition was achieved by altering the maximum applied ΔDY value in conjunction with the simulating time to achieve an initial velocity of 5.23 m/s, as in actual test. The anvil and its contact with the specimen were modeled to incorporate its effect on global behavior of the system. Dynamic tensile properties were implemented using the curve fitting procedures. Separate power law curves were used to include dynamic flow properties over the strain rate range 10^−5^ < ε˙ < 2000 s^−1^. Heat generation due to plastic deformation was not taken into account as the temperature increase was found to be less than 20 °C, when cleavage was initiated.

Banerjee et al. [14] included a model of Charpy test for characterizing the behavior of an armor steel under high strains and strain rates, at elevated temperatures. In their work, model constants of Johnson–Cook constitutive relation [34,35] and damage parameters have been determined experimentally from four types of uniaxial tensile test and they were used a modified constitutive model, introduced as user material sub-routine. There was used a half of the specimen and the impactor, for reducing the computing time. The impactor was modeled as elastic body and the specimen as being made of an elasto-plastic material. The nodes on the impactor were constrained to move only in the vertical direction, impactor velocity being as the initial condition. Johnson–Cook failure criterion was applied only to a narrow region including the notch, along the specimen height. The constants of constitutive and failure model were deduced from tensile tests. This simulation used explicit time integration and was run for a total time of 2 × 10^−3^ s. The simulation results are validated by experimental ones, those being in reasonable agreement.

Recent tests provided complex data on the behavior of polymeric materials, with increasingly complex constitutive models, which take into account several factors: Temperature, deformation rate, structure, and stress.

The importance of modeling the behavior of the material, regardless of its nature, is emphasized by valuable works [23,34,35,36]. Recent studies, experimental and obtained by simulation, have emphasized the importance of modeling the material at high deformation rate, [22,37,38,39,40]. However, most studies refer to metal alloys, used in aero, space and military applications. Equipment for testing materials at high deformation rates does not reproduce the conditions encountered in practice and the obtained results, although realistic, cannot be extrapolated to actual applications without accepting a higher degree of risk than in the case of low deformation rates. Hence, the usefulness of numerical simulations, along with experiments, presents an alternative for studying of damage processes.

As the temperature increases, yielding may occur and its limits decreases with increasing temperature. The deformation at break, increases with increasing temperature, the polymer becoming a brittle material at low temperature, but a ductile material at higher temperature. The effect of the loading rate on the stress–strain curve is opposite to that caused by temperature. At low loading rates, the polymer may behave ductile, with a more pronounced hardening but at high loading rates, the same polymer behaves as a more brittle material.

For polymeric materials, simulation of Charpy tests are few, but the interest in using them as shock absorbers and protective materials makes this subject to be the treated in this study.

## 2. Materials and Constitutive Models

Samples were obtained by injection molding at Monofil Savinesti SA, Romania, the compositions being given in Table 1 [41,42]. Details of the laboratory scale process for obtaining them are given in [27].

Based on the documentation [43,44,45,46], the authors modeled the four materials with the help of multilinear elasto-plastic curves. The results reported in the literature show that polymers also have a stress–strain curve dependent on the strain rate (if other test parameters are kept constant) [24,47].

The tested polymeric blends are based on polyamide 6 (PA6) (Radicci, Bedeschi, Italy), polypropylene (PP) (Sabic, Riyadh, Saudi Arabia) and EPDM (ethylene propylene diene monomer rubber) (Exxon Mobile Chemical, Houston, TX, USA) Polybond 3200 (SI Group, Schenectady, NY, USA) and Kritilen PP940 (Romcolor 2000, Bucharest, Romania), the blend concentration being given in Table 1. 

The specimens were tested on the INSTRON 2736-004 tensile test machine (Instron^®^, Norwood, OH, USA) (from Advanced Materials Laboratory, INCAS, Bucharest, Romania), with a test speed of 1000 mm/min. The specimens are of type 1A, in the shape of a dumbbell, according to SR EN ISO 527-2.

The material model was based on experimental data obtained from traction, for a speed of v = 1000 mm/min [27] because it was estimated that, for these materials, the influence on the shape of the stress–strain curve of the test rate, except for very small test strain rate (v = 10 mm/min), is low and the characteristic values obtained from the tensile tests are in a narrow range. For instance, Musteață [42] reported small differences in strain-stress curves, obtained from tensile tests for PA6 and PP, under test rate of 250–1000 mm/min (approx. 35 × 10^−3^ s^−1^ to 150 × 10^−3^ s^−1^), but different shapes and characteristic values for the test rate of 10 mm/min (approx. 10^−3^ s^−1^). Charpy test with 1 m/s could have 10–25 s^−1^, taking into account the possible span between anvils. 

## 3. The Model

For this model, the impactor and the anvils were considered as rigid bodies as the differences in material properties between those and tested materials are more than one order of magnitude (around 60, considering the tensile strength limit). Analyzing the fracture in this model, for all materials, one may notice that, even if, at crack initiation, a symmetry of the stress and strain fields is visible, this is disturbed by the occurring of high stress and strain. To use the whole sample volume in the simulation seems to be more reasonable for having a more realistic model.

The issue of the mesh network for the Charpy model is the big difference between the dimensions of the specimen and the dimensions of the notch. Choosing a uniform mesh network would increase the time required for the simulation, without bringing significant improvements in the results of interest (stress and strain distributions, etc.). In [16,18,19,20,21], there are given several solutions of mesh networks for modeling and simulating the Charpy test.

The model parts are presented in Figure 1a, including two supports, the notched specimen and the impactor. The mesh network was selected to simulate the Charpy test as presented in Figure 1b. The size of the sides of the elements is between 0.25 mm and 0.75 mm and the number of nodes and elements for each part of the test system is given in Table 2. Growth ratio between elements 1.2, relatively lower than in the discretization of other works [17]. In addition, the model was formed entirely, because in the first simulations with a coarser mesh, it was observed that, although the application of the load and the geometry of the model are symmetrical to a vertical plane, stress distributions and the tendency to zigzag under the notch appeared, a fact also observed on the surfaces of broken specimens.

From experimental results, the four materials have different characteristics, including values for EPS (equivalent plastic strain) at break (Table 3).

Equivalent plastic strain at break (EPS) for each material was calculated as shown in Figure 2, on the curve of each material, by extracting the elastic strain corresponding to the proportionality line of the curve from the total strain at break. The experimental curve for the constitutive model of each material was selected by the authors from those in Figure 3, right column, considered as typical.

Figure 3 presents the results from tensile tests (first column of plots), the selected curve for introducing as constitutive model (second column) and the ten-point true stress–true strain curve used as constitutive model for each material. The test results in the first column are expressed using tensile stress (engineering),$\;\sigma_{eng}$, and tensile strain (engineering), $\varepsilon_{eng}$*,* calculated by means of the relations:(1)$$\sigma_{eng}=F/A_{0}$$
(2)$$\varepsilon_{eng}=\Delta L/L_{0}$$
where *F* is force applied at time t, *A_0_*—initial cross area of the specimen, *L_0_*—initial length of specimen between marks, *ΔL*—the change in length between marks, at moment *t*. 

If the engineering values, $\sigma_{eng}$ and $\varepsilon_{eng}$, are known, the relations used to calculate the true strain, $\varepsilon_{true}$, and true stress, $\sigma_{true}$, are: (3)$$\varepsilon_{true}=\ln\;\left(1+\varepsilon_{eng}\right)$$
(4)$$\sigma_{true}=\sigma_{eng}\left(1+\varepsilon_{true}\right)\;\varepsilon_{eng}.$$

The assumptions considered in this model are as follows.-The supports and the impactor are considered rigid, given the difference in properties; impactor and rigid supports were also considered in the works [16,18,19,49], but in the work of Sainte Catherine et al. [21] the impactor and the supports were considered perfectly elastic, given the specimens.-The initial velocity (just before impact) is v_0_ = 0.96 m/s (impact velocity, with which the actual tests were performed).-Boundary conditions: Friction between specimen and supports, friction between specimen and impactor (COF = 0.3) for all analyzed cases; Poussard et al. [19], Haušild [17], and Sainte Catherine [21] did not consider friction, but the friction between impactor and sample and supports was taken into account in the models from [42].-Failure criterion was selected for equivalent plastic strain at break (EPS): When the critical value of EPS at break is reached in the model, the crack is initiated.

From the experimental results, it was found that the four materials for which the impact behavior was simulated with an impactor having linear trajectory, have different values for EPS at break, as calculated from the diagram true stress–true strain (see Table 3).

## 4. Experimental Results of Charpy Tests

In this paper, the tests were performed on the CEAST 9340 impact test machine (Instron^®^, Norwood, MA, USA) within the materials strength laboratory at “Politehnica” University of Bucharest. The tests were performed with a hemispherical impactor, with a speed of 0.96 m/s and the mass of the impactor 3.219 kg. The specimen has the drawing shown in Figure 4 and Figure 5 shows the geometry of the impactor.

According to the SR ISO 179 standard, ten tests were chosen for each material, considered to be typical for the respective material.

Figure 6, Figure 7, Figure 8 and Figure 9 present two characteristics: The force and the absorbed energy as functions of time.

The very short time to failure shows the more fragile nature of both PP and PPm (Figure 6). The addition of Kritilen PP940 in PP does not introduce noticeable changes in the shape of force–time and energy–time curves. For the material PPm, the force–time curves have large oscillations especially in the first part of the destruction process, the oscillations being of 30–60 N, which means a mixed structure of amorphous and crystalline volumes. 

The breaking time for most of the specimens wqs up to 1 × 10^−3^ s except for test eleven, for which the specimen broke at t = 0.9 × 10^−3^ s. On absorbed energy–time zones, two zones can be distinguished. The first zone is up to 0.07 J, having a very large curvature, followed by an almost linear zone. Except for test 11, the results fall in a very narrow band of 0.025 J. In the case of PPm, several specimens were broken at higher time values, considered for F = 0. It is noticed that PPm has slightly higher values than those obtained for PP, both for F_max_ and energy absorbed in impact (Table 4). The time to failure (F = 0) is approximately the same for PP and PPm (0.85–1.4 m/s). The addition of this agent Kritilen PP940 has improved the behavior of PP on impact, even if its concentration is only 1% and it also improved the shape quality of the samples when they were extracted from the mold. One may compare the curves for the neat polypropylene to those for PP + 1% Kritilen PP940, coded PPm.

The behavior of material H is similar to that of the material PPm (Figure 7) and the time to failure (considered at F = 0) is close, from 0.9 × 10^−3^ s to 1.29 × 10^−3^ s. The oscillations of the force–time curve show the presence of two polymeric structures, one more tenacious and the other weaker. The oscillations of the force–time curve show that in the breaking section the material is composed of an alternating succession of more resistant micro-volumes and weaker materials from a mechanical point of view, but also amorphous and crystalline micro-volumes of the two constituent polymers (PP and PA6). 

Material G (Figure 8) has much smaller amplitude of force oscillations. The oscillations start with values between 30–60 N, after which their amplitude does not exceed ~20 N. The maximum values of the force are concentrated between 150–180 N, which reflects that the specimens are more similar in terms of response to impact, that is the quality of the injection process is more controllable for this material. The energy–time curves are very close and very similar. The scattering band of the curves is narrower, up to 0.5 J (up to 1 ms in time) and then on the linear area it is slightly more scattered. The value of the moment of complete failure (F = 0) is in the range of t = 2.16 × 10^−3^ s to t = 3.77 × 10^−3^ s. If material G is compared to the other materials (see Table 4), it is observed that the maximum value of time to failure has the value of t = 3.77 × 10^−3^ s, the highest of all materials. If the materials are ordered in descending order, from the greatest value of time to failure (F = 0), this is G, PA6m, PA6, PP and very close to each other, PPm and H. For F_max_, the highest value is for PA6, followed by PA6m.

A designer will be interested in values for absorbed energy at break and, the blend PA6 + EPDM (material with the code PA6m) has values higher than all tested materials, including PA6, a polymer that is used in applications for its impact resistance (Figure 9). Force–time curves for both materials are similar but the time till break is longer for PA6m. The energy at break plot for PA6m is characterized by two zones: A curved line in which the energy increases after a curve up to 0.15 J, after which the evolution of the energy stored in the test tube is linear.

The SEM images (Figure 10) shows that the introduction of PA6 in higher concentration changed the morphology of the blend. Material H (Figure 10a) has a PP matrix with PA6 droplets and pores of the order of a few microns. The matrix has a brittle fracture. The droplets located on the breaking zone are very deformed, elongated, and the matrix breaks as a more brittle polymer, with linear micro-flows, probably as a result of a lower crystallinity of PP. Material G (Figure 10b) has the inverted phases, as it resulted from the EDX analysis. The PP droplets are larger and do not have cohesion on the whole surface with the matrix. 

Analyzing both SEM images for material PA6m (Figure 10c,d), this blend of PA6 and EPDM seems to be a homogeneous one: (a) A low magnification of the arrested crack and (b) a detail that point out that the blend is homogenous and extensively deformed.

The addition of EPDM in PA6 produces a force–time diagram similar to that of simple polyamide, but the failure occurs after a longer time interval. At t = 2.59 × 10^−3^ s, for PA6, all the specimens were already broken, whereas for PA6m the damage of the specimens (with incomplete failure), is achieved between t = 3.34 × 10^−3^ s and t = 3.59 × 10^−3^ s. In other words, the addition of EPDM increases the time to failure (even partially, in the case of PA6m material). The evolution over time of the destructive energy has similar forms only that most of the PA6m specimens break partially after storing an energy between 0.305 J and 0.389 J, while PA6 reaches only 0.279 J at the complete failure of the specimens.

Comparing the four materials (PPm, H, G and PA6m), the value of the maximum force was obtained for PA6m (Figure 11). The material for which the following maximum force value was obtained is G, with 170.7 N, only 6.46% lower than the PA6m material. PPm has the maximum force about half of that for G, and the lowest value was obtained for material H, in other words, this is the weakest material in terms of the maximum force required for damage.

Regarding the energy at break (also named absorbed energy till break) (Figure 12), it is noticed that the best results were obtained for PA6m, both for the average value of the energy at break (0.344 J) and for the spreading interval, which is below 0.084 J. PPm and H materials have about five times less energy at break than PA6m material. It is clear that in applications with shock resistance materials, PPm and H are not recommended, instead the PA6m (PA6 + EPDM) coat gave better results than for PA6. Material G should not be removed from the recommendations for shock applications with speeds up to 1 m/s, but the scattering of the results is higher and the average value of the energy at break is very close to that of the simple polymer PA6. The choice between these two materials, G and PA6 is made by analyzing other criteria necessary for the design, such as water absorption, dimensional stability, process ability and price.

Aspects of impact-breaking mechanisms at this low speed (0.97 m/s) are shown in SEM images (Figure 13). PPm and H materials have a brittle fracture appearance, while G and PA6m materials have fracture surfaces typical of ductile fracture. All tested PA6m samples showed the same form of failure while maintaining a link between the test fragments (without complete failure) (Figure 13d). Studying these images, it is not possible to specify whether the failure advanced only from the notch or occurred in the impact area of the impactor, also. The aspects of breaking material H are similar to PPm. Material G has a different appearance from the materials discussed above: The breaking surface is much rougher and it is likely that the tear was initiated first near the notch but then under the impactor. The very uneven area is very likely to have yielded more abruptly than the material under the notch and under the impactor, some dispersed droplets are partially detached from the matrix, but others, usually smaller, are trapped in the matrix. The most interesting SEM images were obtained for PA6m (Figure 13d). An area parallel to the breaking surface of the crack is observed in which the material has a pronounced plastic yield at an angle of approximately 45° to the breaking surface. The thickness of this layer is about 200 microns and was not observed in the other materials that contained EPDM (H and G). Stopping the crack progress is also observed in simulation for material PA6m. The addition of EPDM to PA6 led to the propagation of a failure without fibrillation but with strong local flows on a band near the failure surface.

For materials G and PA6m, the results are very good, PA6m and G exceeding PA6 when taking into account the absorbed energy till break. The difference in behavior of more ductile materials (PA6m, G and PA6) and the brittle ones (PPm and H) may be noticed on Figure 14 when force–time curves are overlapped. For material G, EPDM and PA6 formed a matrix more ductile that that in material H and the droplets of PP are better fixed in this matrix. The material PA6m being a blend of PA6 + EPDM, has homogenous aspect and the detail in Figure 10d evidences extensive deformation, but not completely break. Table 4 presents average values of several characteristics of the tested materials, the best values for impact characteristics (absorbed energy at break and impact resistance) being for material PA6m.

The impact strength of the specimen was calculated with the formula
(5)$$R_{impact}=\frac{\Delta E}{A_{0}}\;\left[J/m^{2}\right]$$
where *ΔE* is the energy absorbed up to *F* = 0 and *A_0_ = b(h_specimen_ − h_notch_)* is the transversal minimum section area (at the tip of the notch), *b* being the width of the specimen and *h_specimen_* being the specimen height and *h_notch_* being the depth of the notch.

## 5. Simulation Analysis for Tested Materials

Some images were obtained by considering a half of the transparent specimen (at time t = 0 s) (the specimen is not divided into two entities, and the section is right in the plane of symmetry of the notch). This virtual sectioning facilitates the observation of the development of the failure and the change of the distribution of equivalent stresses. A qualitative and quantitative analysis of stress concentrators and how they evolve at any given time as the failure propagates, can be performed.

Material PPm. At the closest moment to the impact (t = 5.0 × 10^−4^ s), two stress concentrators are observed on the specimen, one under the impactor, the other at the tip of the notch, both presenting a “butterfly” distribution. The higher values of equivalent stress are recorded under the impactor. In the middle of the notch tip, this is below half the stress at break, σ_ech max_ = 13.59 MPa. 

In the following moments, the two stress zones increase in area and value. Thus, at time t = 1.0 × 10^−3^ s, σ_ech_ = 18.2 MPa. At moment t = 1.5 × 10^−3^ s, the crack has not yet been created, but the equivalent stress has increased to σ_ech_ = 23.9 MPa, under the impactor. At t = 2.0 × 10^−3^ s, the equivalent stress increased to 29.6 MPa under the impactor and it was 23.03 MPa at the tip of the notch. The specimen is increasingly stressed at time t = 2.5 × 10^−3^ s (Figure 15a) and the initiation of the crack may be seen in the middle of the tip width of the notch, the crack being already developed on a length of less than 1 mm, the equivalent stress reaching 31.8 MPa. From simulation, until the moment t = 2.5 × 10^−3^ s, the critical value of EPS was not reached, EPS_(PPm)critical_ = 0.09, and therefore, the crack was not initiated.

At time t = 3.0 × 10^−3^ s (Figure 15b), the crack is already initiated and it propagates on the height of the specimen, causing an uneven band of maximum stresses of 33 MPa. The opening of the crack and the continuation of the impact caused a band of equivalent stress almost as big, on the opposite surface, which still does not yet cause the cracking because the impactor causes compression (Figure 15c,d). The middle volume of the specimen between this first crack and the compressed side is almost non-tensioned (in blue and dark blue colors), with values of 3–7 MPa.

At t = 3.5 × 10^−3^ s (Figure 15c), the crack is already propagated over almost half the distance between the notch tip and the impacted surface. The highest equivalent stress is observed under the impactor σ_ech_ = 38 MPa. At time t = 4.0 × 10^−3^ s, the propagation of the second crack is observed from the impacted part of the specimen to its center. The maximum equivalent stress is visible only at the tip of one of the cracks, at that one initiated from the notch tip. On the specimen height, the still undetached area appears to be 2 mm. At this moment, the material left between the two cracks has only a small area, with low stresses of 7 MPa. 

Stress concentrators are created at the edges of the theoretical impact line due to the fact that large deformations appear as an edge effect at the ends of the impact line (Figure 16). This virtual cross section allows for pointing out that the crack is initiated at t = 2.5 × 10^−3^ s, in the middle of the notch width (Figure 16a). Figure 16b,c show how the crack initiated from the notch tip advances. At time t = 4.5 × 10^−3^ s (Figure 16d), the specimen has two broken areas, the one propagated from the notch and a smaller one under the impactor. A maximum equivalent stress of 32 MPa is noticed, in a narrow ribbon remained between the tips of the two cracks. Although at the beginning, the impact had symmetrical characteristics as compared to the initial plane passing through the notch tip, it is observed that, later, the stress distribution within the material becomes slightly asymmetric. This phenomenon can occur due to the automatic discretization of the model, although a fine mesh was used and also to the elasto-plastic behavior of the material. 

At time t = 5.0 × 10^−3^ s, the still unbroken material has a size of less 1 mm, this decreasing at time t = 5.5 × 10^−3^ s, leading to the growth of the two cracks, the crack starting from the notch tip being longer than that started from the impacted surface. Linkages (small bridges) between the two fragments of the specimen, distributed unevenly across the width of the specimen can be observed. At moment t = 6.0 × 10^−3^ s, the specimen fragments are already separated. If high equivalent stresses occur at later times, they are due to the pushing of the fragments into each other until the impactor separates them. The maximum equivalent stress moved to the tip of the crack started under the impactor and has a value of 33 MPa. The moment t = 6.5 × 10^−3^ s is characterized by σ_ech max_ lower than the previous one, meaning that the failure process alternates with a process of material relaxation (σ_ech_ decreases), which is specific to plastics. At t = 7.0 × 10^−3^ s, σ_ech_ = 33.29 MPa and that may be seen in the virtual cross section of the specimen, which means that there are still small areas unbroken (unseparated), local bridges between the two already detached fragments of the specimen. At time t = 7.5 × 10^−3^ s, only the crack under the impactor may be stressed due to compression, due to the opening of the specimen. Starting from t = 7.5 × 10^−3^ s till t = 1.0 × 10^−2^ s, the equivalent stress drops to 20 MPa. These stress values, without reaching the stress value at break, are due to the collision of the two fragments of the specimen. 

The smooth surfaces that appear from the moment t = 7.5 × 10^−3^ s, indicate the areas where the upper crack is closed due to the rapid movement of the fragmented gates of the specimen.

The conclusion drawn from the above comment is that the impact does not generate high stress in all the volume of the impacted specimen, but determines local volumes of stress concentrators where the crack starts.

Figure 17 presents a selection of moments during the impact simulation for material H, von Mises stress distribution being better observed in lateral view, the appearance and development of the crack at the notch tip and then of the crack under the impactor. Due to colored scales, the areas with maximum von Mises stress values (red on each image) are well highlighted. The end of the simulation for material H is at the moment t = 5.0 × 10^−3^ s. 

In the details in Figure 18, particularities of the breaking process may be noticed: (a) Crack initiated at t = 2.0 × 10^−3^ s, (b) at time t = 2.5 × 10^−3^ s, the two cracks advances, less of 1/3 of the specimen height being unbroken and (c) this virtual cross section reveals the asymmetric stress distribution at the cracks’ tips. The failure propagates from the notch (first) and then from under the impactor. There is an area destroyed by crushing (compression), just below the impactor, as seen in SEM images. The distribution of equivalent stresses at the tip of the cracks is typical of visco-plastic materials. The perfectly smooth surface visible in Figure 14a is virtually resulted by sectioning with the imaginary plane. It is observed how the crack advances from one moment to another and the area where the last strip of material is lasting (t = 3.25 × 10^−3^ s) before the total fragmentation of the specimen.

If a qualitative comparison is made with SEM images of the H material, they are close enough in appearance. In Figure 18, one may notice that the material H has a fragile failure and the initiation of the crack is made first from the notch tip, but there is a second crack, initiated from under the impactor, which joins the first at almost 1/4 of the height of the specimen. The actual appearance of the broken surface (Figure 19) resembles the aspect of the failure at the end of the simulation, but the crack under the impactor is less developed on actual specimen and a superficial crushing destruction is observed, which could not be highlighted by modeling. The cracking occurred on a shorter length from the impactor.

From experimental data, material G is the second material, in a ranking based on energy at break (see Figure 12). For this reason, the simulation of impact failure is important when compared to the material with the best results, PA6m. Figure 20 shows the evolution of the crack generated from the notch tip. 

Figure 20a presents the initiation of the crack at the notch tip, the zig-zag development of the crack from the notch tip (Figure 16b–d), the generation of the second crack under the impactor, starting from both edges of the specimen (Figure 16e) and a moment when a small material bridge is remained between the two fragments (Figure 16f), until the complete separation of the fragments of the specimen (see moments in Figure 21). Comparing the images in the simulation from one moment to the next one allows for establishing the start points of the cracks and how they develop.

For material G, in Figure 20 there are given several moments during the impact: (a) The impactor has created on the impact surface an equivalent stress field of high values, close to the stress at break and deformed the specimen in order to force the development of the crack from the notch tip, (b) and (c) present the same moment but from different angles, the first pointing out the high stress field under the impactor, the second making visible the zig-zag aspect of the crack, (d) crack is developing and the specimen is bending under the impactor motion, (e) the initiation of lateral two cracks on the specimen width under the impactor, (e) this crack having a slower propagation speed, in the middle of the specimen maintaining a high stress field. Lateral views of the samples in Figure 21 presents the crack just before complete separation of the fragments. The impactor pushes the specimen, making the fragments to change their positions from horizontal to oblique ones and it is observed that the union of the two cracks suddenly diminished the stress field in the area, strong stress concentrators being in the material ribbon (bridge) left unbroken (a and b) and at t = 1.5 × 10^−2^ s, the resulted fragments are completely separated (c).

Simultaneous development of stress concentrators is noticed, one around the notch tip and the other under the impactor, but the failure criterion (EPS = 0.156) is reached for the first time at the tip of the notch, at time t = 3.75 × 10^−3^ s, the zone of high von Mises stress (with red color) remaining at the tip of the crack, advancing as it grows, but there is also an area of the same values under the impactor, as the impactor pushes the specimen.

The opening of the crack and the change of the angle between the still undetached fragments of the specimen concentrate more the area with high stress. 

Figure 22 presents important moments during simulation of impact on material G, in a virtual cross section of the specimen: (a) At the beginning of the impact (t = 7.5 × 10^−4^ s), it is observed the formation of two zones of stress concentration on the width of the specimen, along the notch tip and under the impactor, but the crack is not yet initiated, (b) the failure occurs between moments t = 3 × 10^−3^ s and t = 3.75 × 10^−3^ s, and the crack begins and develops in the middle of the notch tip width (c) at time t = 4.5 × 10^−3^ s, the crack is propagated over the entire width of the specimen and has already advanced by almost 1.5 mm, (d) the crack advances towards the opposite surface of the specimen and the von Mises stress distribution has high values on the impacted surface. At time t = 6 × 10^−3^ s, the specimen is not yet completely broken, leaving a band of ~2 mm still uncracked. Another step in the crack development is shown in Figure 16f, showing a moment of crack converging for material G.

Comparing the image in Figure 22d with SEM image (Figure 13c), one may notice that:-The aspect of failure is rough, both in the model and on the actual specimen, the actual uneven aspect of the specimen surface being probably caused by the inhomogeneity of the material; and-a very thin compressed strip is observed on the specimen side the impactor stroke.

A virtual cross section of the specimen made of material G reveals how the cracks are generated and the shape change of the specimen: (a) and (b) the crack initiates between t = 6.25–7.5 × 10^−3^ s, being located in the middle of the notch tip, (c) at t = 1.125 × 10^−2^ s, this advances quickly with larger stress concentration zones in front of the crack tip and on the impacted side and the second crack, under the impactor is initiated, (d) the cracks advance intermittently (one is developing faster, the other is waiting to act and then the roles are changed).

The breaking process is continued and the maximum values of von Mises stresses remain between 47.8 MPa and 43.74 MPa, which means that at any time between t = 1.375 × 10^−2^ s and t = 2.5 × 10^−2^ s, the material is still breaking. 

In Figure 23, several successive images were chosen to show how the crack advances in the most ductile material (PA6m): (a) high values of von Mises stress are visible in a butterfly band around the notch tip and under impactor, (b) the crack was started from the tip of the notch, in the middle of its width at t = 8.75 × 10^−3^ s, (c) second crack is visible, under the impactor, (d) first crack advances more quickly and the volume between cracks are highly stressed, (e) the remaining band of material between the cracks is very stressed and the surfaces of the second crack are pushed one against the other due to the movement of the impactor and the orientation as a letter V of the specimen, and (f) at the end of the simulation only a stressed spot is seen and there is obvious that the two fragments of specimen are not completely separated. 

From the sequence of moments in failure process in Figure 23, it can be seen that:
-the specimen did not detach completely (both experimentally and in simulation),-the running time is the longest for this material, as compared to the other modeled materials, and-for PA6m material, the EPS at break has the biggest value.

The area under the impactor was strongly deformed, observing the developing trapezoidal shape, with the larger side under the impactor (Figure 24).

Analyzing the actual aspect after specimens had been tested (Figure 25), it was observed that: (a) material PPm had a brittle fracture and (b) material PA6m is more prone to local deformation, having a very narrow band of material that did not break, as noticed in the simulation.

Especially after crack(s) initiation, the aspect of stress distribution is not perfectly symmetric, the explanation being the more viscous-plastic character (larger plastic deformations and low slope in the plastic field). The propagation of the crack under the impactor, along the entire width of the specimen, relaxes this area, the stress concentrator arching between the two cracks.

Qualitative validation of the crack propagation mode in the PA6m material may be argued by comparing the actual shape of the cracks (Figure 25b) with the simulated shape (Figure 23f).

## 6. Conclusions from Comparing Simulation and Experimental Results 

The review of available and recent impact modeling documentation for Charpy test highlighted mesh pattern too coarse or too fine, disregard for friction in the majority of works, simplified modeling of specimen material, with only a quarter or a half of the test system (specimen, impactor and supports) and simplifying hypotheses that may alter the virtual behavior of the specimen.

Moments when the crack initiation is visible on the simulation are presented in Figure 26. For material H, the crack appears at time t = 2 × 10^−3^ s, for PPm, this appeared at time t = 2.5 × 10^−3^ s. Material G has the crack initiated at time t = 4.5 × 10^−3^ s. The longest time was recorded on the simulation for PA6m, t = 2.5 × 10^−2^ s. From these images, it may be noticed that materials PPm and H are less deformable, the deformation of the notch shape being almost imperceptible. For more ductile materials, the V of the notch opens visibly and even the crack shape differs for G and PA6m. For material G, the crack propagates of brittle type, it develops towards the impactor. For material PA6m, a strong deformation of the material without the crack advancing so rapidly, this is also visible on SEM images from the same angle (see Figure 13d).

The fracture of the model is uniform on the sample width and it advances uniformly in the section. The actual fracture of G and (even PPm somehow) has o core with brittle aspect and ductile aspect around. This could be explained by the fact that material model does not take into account the influence of strain rate and non-homogeneity of amorphous and crystalline micro-volumes. 

FE simulation of Charpy tests for different polymeric materials with a material constitutive model deduced from tensile tests has produced results matching experimental data, validating the model constants for low impact velocity (1 m/s). 

Table 5 points out the important moment in the failure process but also the moments they occurred. Differences point out the more ductile character of the materials G and PA6m and that more fragile for materials PPm and H.

Figure 27 presents the evolution of three parameters during the impact simulation: (a) The maximum values of the von Mises stress for the four material models, (b) the evolution of the impactor velocity during the impact, and (c) the evolution of deceleration of the impactor. 

In general, the maximum values of von Mises stresses indicate the initiation of a crack and/or its continuation after a very short moment of relaxation of the stresses. The last maximum value can also indicate a strong, very punctual compression of the specimen fragments, if they collide. Analyzing the images, it is likely that the last maximum of material G is due to this compression contact. PPm and H materials have graphs of the maximum values of von Mises stresses with distinct peaks (three for PPm and two for H), indicating cracks (one starting from the notch and the other initiated under the impactor) and/or hits of the fragments with strong local compression. For G and PA6m materials, these curves reflect a much more ductile behavior. Analyzing the plots for maximum value of von Mises stress, during a time interval at 3 × 10^−2^ s, the materials could be in two behavioral groups:-materials with brittle fracture (H and PPm),-materials with ductile fracture (G and PA6m).

If it is analyzed the time until the breaking of the Charpy specimen, it is observed that this time increases from material H, PPm, G, to PA6m, for which the breaking is not with the total detachment of specimen fragments (Table 5). In reality, the time till failure of Charpy specimens is ranked in the same way as in simulations (see Figure 6, Figure 7, Figure 8 and Figure 9).

The values of residual velocity, *v_res_*, (considered the impactor velocity at the end of simulation) are in the same order as the EPS values. This value of residual velocity was used to calculate the absorbed energy for the simulation, *ΔE_model_*, for each material:(6)$$\Delta E_{model}=m_{impactor}\left(\frac{v_{0}^{2}}{2}-\frac{v_{res}^{2}}{2}\right)\;\left[J\right]$$
where *m_impactor_* is the mass of the impactor, *v_0_* is the initial velocity of the impactor.

The maximum value of deceleration, a_max_, was used to calculate the maximum value of the force striking the specimen, *F_max_*, during the simulation, in order to compare it with that experimentally obtained for the same material:(7)$$F_{max}=m_{impactor}\times a_{max}\;\left[N\right]$$
where *m_impactor_* is the mass of the impactor, *v_0_* is the initial velocity of the impactor.

For the material constitutive models introduced in the simulation, the lowest value of EPS produced the smallest reduction of the impactor velocity; as the residual velocity enters the formula of the energy absorbed by the test specimen, it results that the material with the highest EPS (PA6m) absorbed the greatest energy. The conclusion would be that the residual speed is proportional to the EPS value (Table 6).

Small differences in energy at break and residual velocity were obtained between the model and actual material parameters for materials PPm, H and G (Figure 28c). The best concordance for these two parameters was obtained for material H, followed by G and PPm.

Comparing the time till break for true and simulated tests (Figure 28a), the FE model exhibits greater values than the experimental ones. PPm and PA6m have larger gaps for this parameter, but for materials H and G, this is lower. True time till break is taken as the greater time value for F = 0 on force–time curves, for each material. When comparing maximum force (Figure 28b), real and model values are close for material PA6m, for the other materials, the difference being constant, but with same trend. The residual velocity and the absorbed energy are parameters with close evolution of the values for model and real data for materials PPm, H and G, the bigger difference being obtained for material PA6m, meaning that the strain rate and the temperature influence more the behavior of ductile polymer blends.

The biggest difference between the model and the actual test, taking into account these parameters, was obtained for PA6m, the material with better impact characteristics. From this analysis, it results that the constitutive model of multilinear material introduced in the simulation of the Charpy test gave results closer to reality for the more fragile materials. For the most ductile material (PA6m), with a high yield value, the model had a much higher impact energy absorption than the actual one, which would mean that the introduction of the multilinear model obtained at a test rate of 1000 mm/min (0.016 m/s) from tensile tests, does not satisfactorily simulate its behavior, at impact with speed v = 0.96 m/s, being known that even in the field of low impact velocities (1–10 m/s), polymers are sensitive to the rate of stress and implicitly of deformation rate, usually in the sense of increasing the stress at break and decreasing the strain at break. Also, differences could be explained by the isothermal characteristic of the model during simulation, the actual impact generating local heating as results of transforming a part of the absorbed energy into heat.

The originality of the proposed model consists in:-Finer discretization of the impact area to highlight the mechanisms of failure and their development in time;-this made it possible to differentiate the destruction mechanisms according to the material introduced in the simulation;-in the simulations were introduced simplified curves (in 10 points) of the most representative curves real stress-real deformation for the materials of the mixture family;-depending on the base polymer (PA6 or PP), the initiation and development of failure for the 4 materials were highlighted;-the author used a less widely used destruction criterion, EPS;-the validation of the model and the simulation results was done;-qualitative: the shape of the broken surfaces by comparing to SEM images; and-quantitatively by comparing the impact duration: The duration of the destruction of the specimen is longer than the real one, explainable by the fact that the material model does not take into account the influence of the material deformation speed in the Charpy test, the model being introduced with a 0.016 m/s (1000 mm/min) (maximum strain rate for tensile tests). It is very likely that the appearance of the stress–strain curve will change for 1 m/s in the sense of increasing the yield limit and decreasing the EPS. The higher the value of EPS and the higher the yield point, the wider the distribution of high stresses in a larger volume.

## Figures and Tables

**Figure 1 materials-13-05837-f001:**
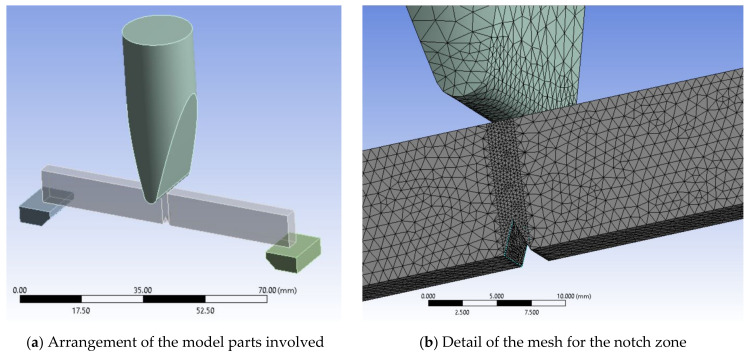
Mesh network of the model.

**Figure 2 materials-13-05837-f002:**
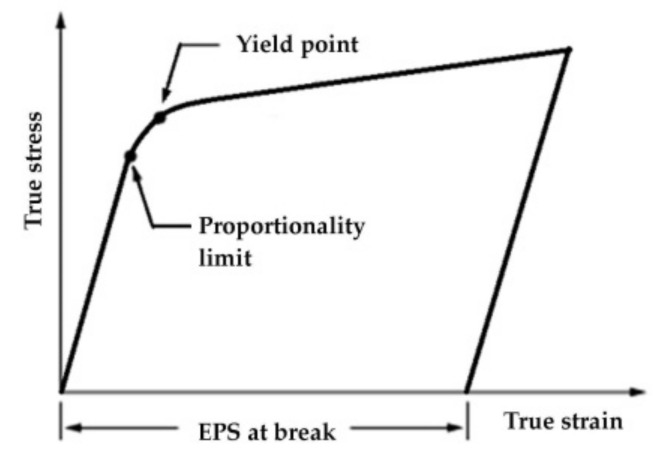
Equivalent plastic strain (EPS at break), on true stress–true strain curve [48].

**Figure 3 materials-13-05837-f003:**
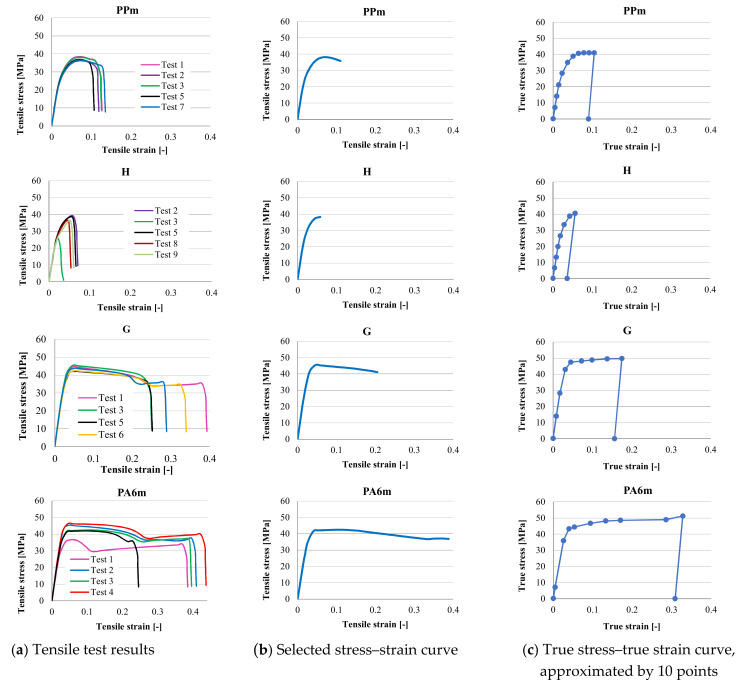
Tensile tests and models for the materials.

**Figure 4 materials-13-05837-f004:**
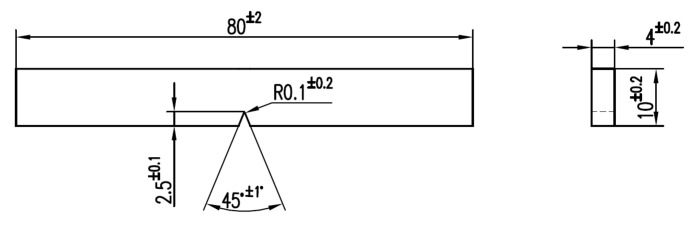
Specimen dimensions (dimensions in mm, angles in degrees).

**Figure 5 materials-13-05837-f005:**
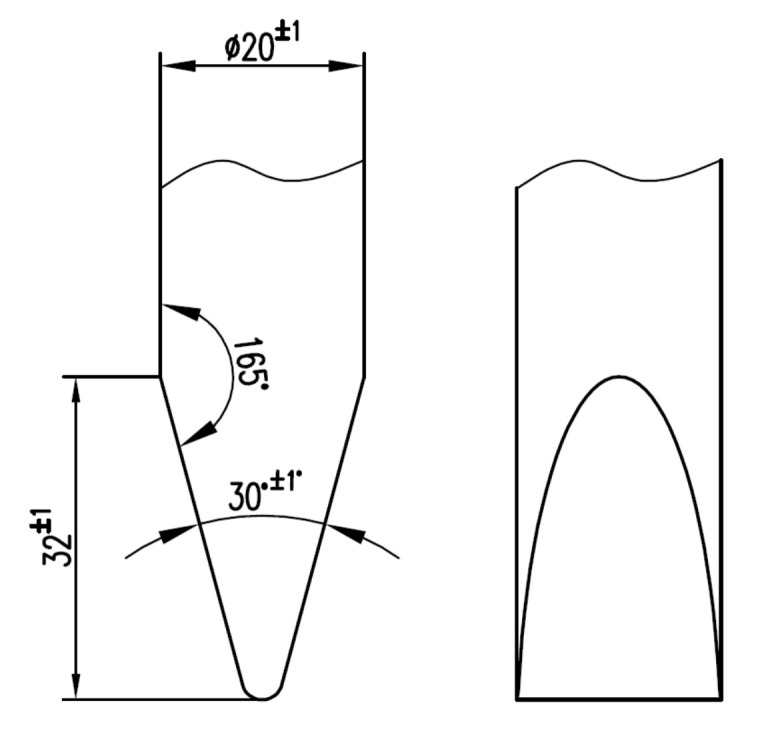
Impactor geometry (dimensions in mm, angles in degrees).

**Figure 6 materials-13-05837-f006:**
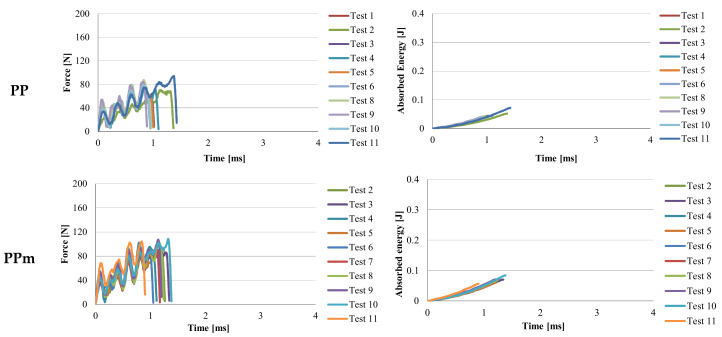
Curves for force and absorbed energy in time, for materials PP and PPm.

**Figure 7 materials-13-05837-f007:**
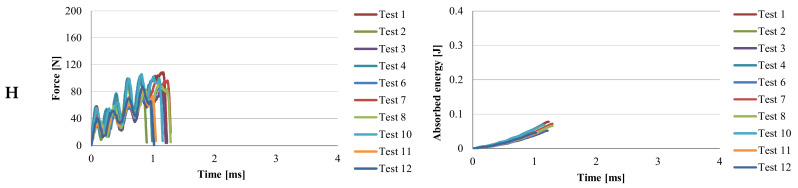
Curves for force and absorbed energy in time, for material H.

**Figure 8 materials-13-05837-f008:**
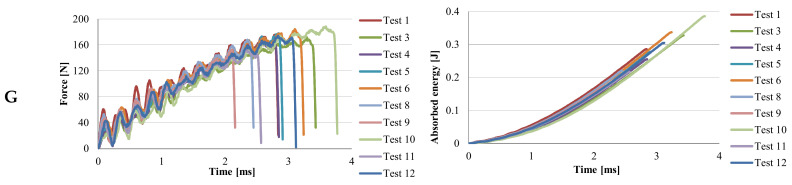
Curves for force and absorbed energy in time, for material G.

**Figure 9 materials-13-05837-f009:**
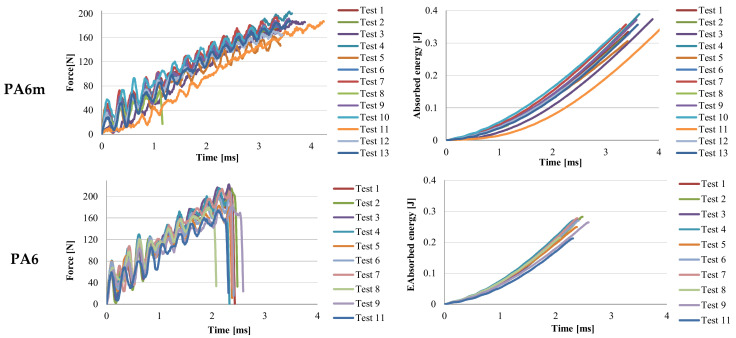
Curves for force and absorbed energy in time, for the tested materials.

**Figure 10 materials-13-05837-f010:**
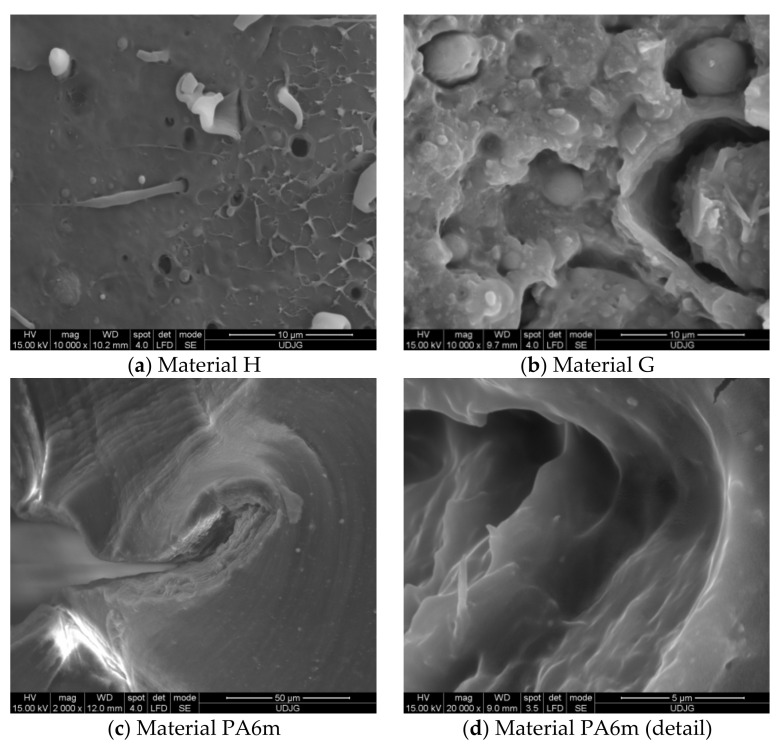
Fracture morphology, obtained for Charpy test.

**Figure 11 materials-13-05837-f011:**
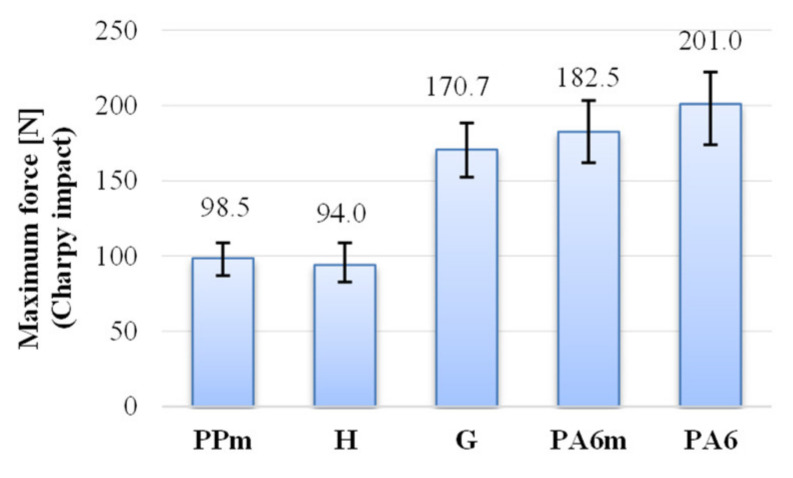
Average values of the maximum force.

**Figure 12 materials-13-05837-f012:**
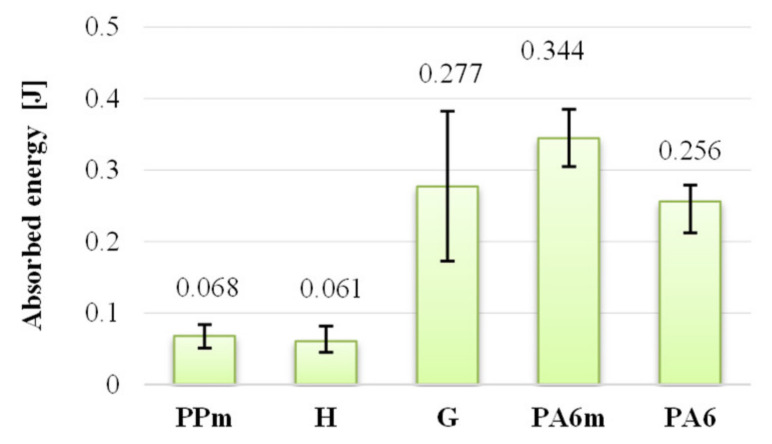
Average values of the absorbed energy.

**Figure 13 materials-13-05837-f013:**
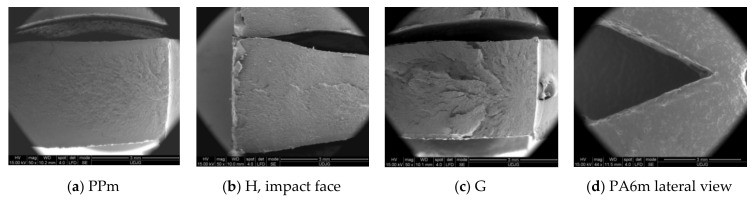
Typical aspects of the surfaces obtained after sample breaking.

**Figure 14 materials-13-05837-f014:**
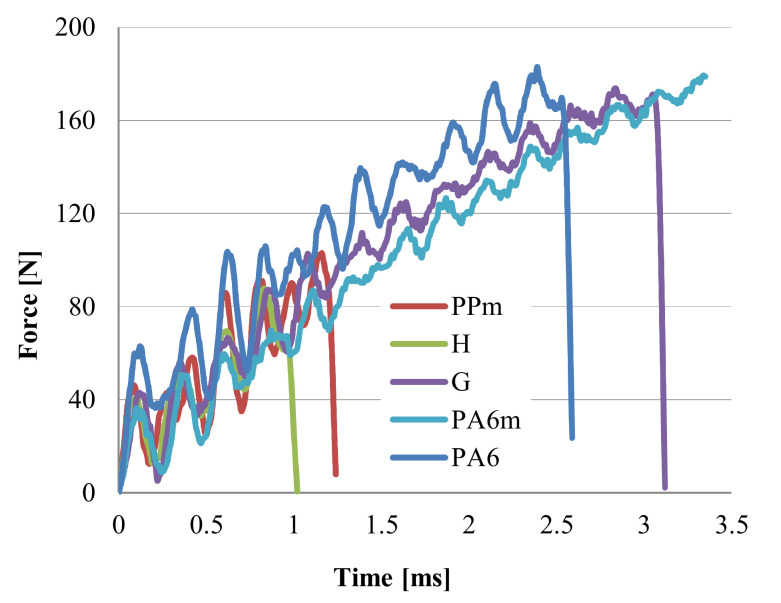
Typical curves force–time for the tested materials.

**Figure 15 materials-13-05837-f015:**
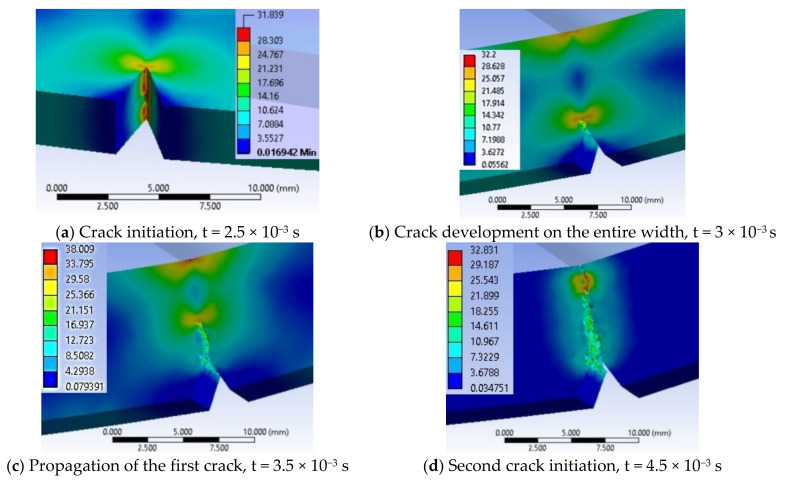
Selection of important moments in simulation for material PPm.

**Figure 16 materials-13-05837-f016:**
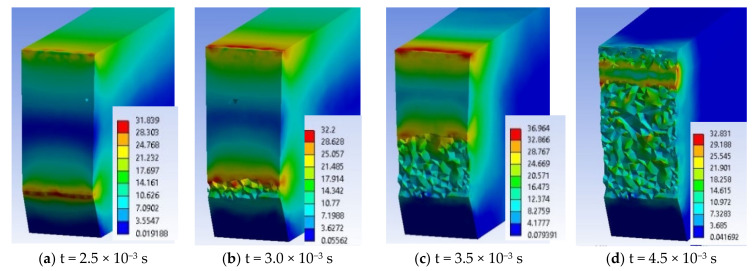
Successive moments from the impact of the test specimen made of PPm material.

**Figure 17 materials-13-05837-f017:**
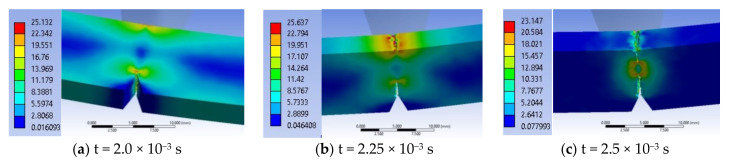
Selection of important moments in simulation for material H (equivalent stresses, MPa).

**Figure 18 materials-13-05837-f018:**
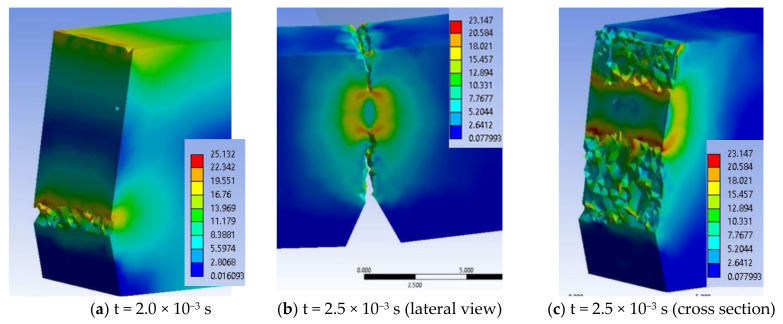
Failure details, for material H.

**Figure 19 materials-13-05837-f019:**
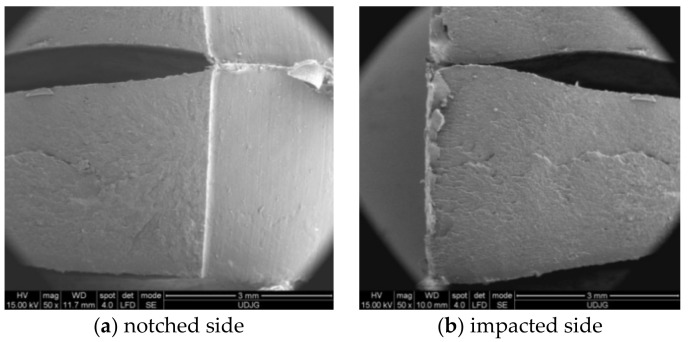
The failure surface of a Charpy specimen, made of material H.

**Figure 20 materials-13-05837-f020:**
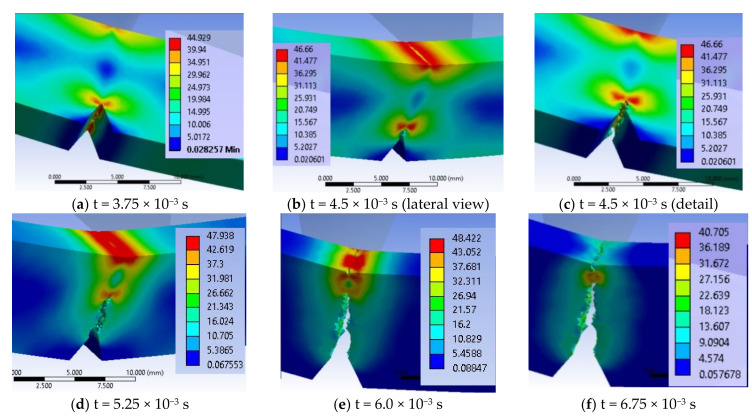
Evolution of the crack generated from the notch tip, for material G.

**Figure 21 materials-13-05837-f021:**
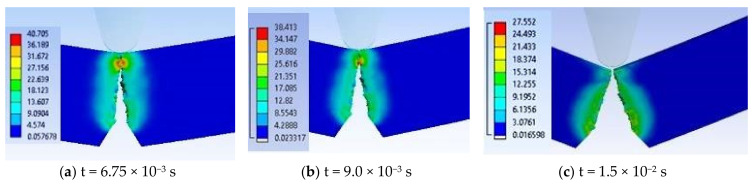
Crack evolution under the impactor, until the separation of the broken fragments, for material G.

**Figure 22 materials-13-05837-f022:**
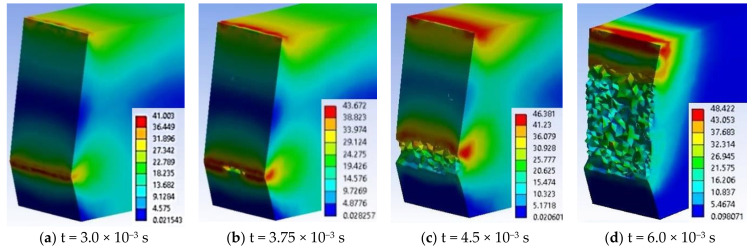
Evolution of the crack generated from the notch tip, before the crack initiation under the impactor, for material G (von Mises stress distributions in MPa).

**Figure 23 materials-13-05837-f023:**
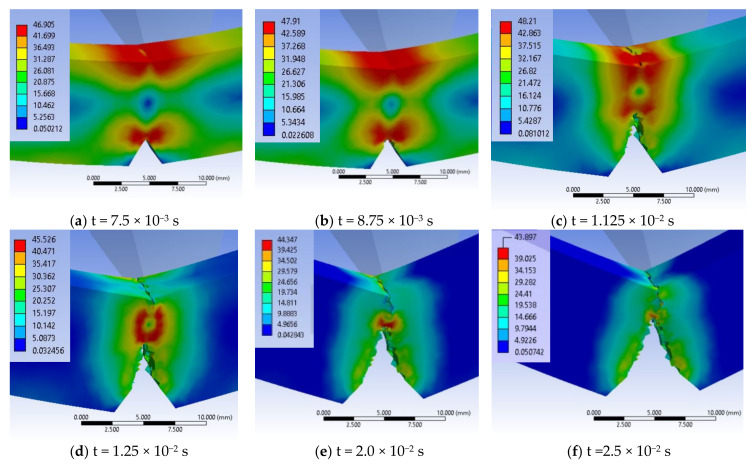
Lateral view of specimen made of PA6m, during impact, with von Mises stress distributions in MPa.

**Figure 24 materials-13-05837-f024:**
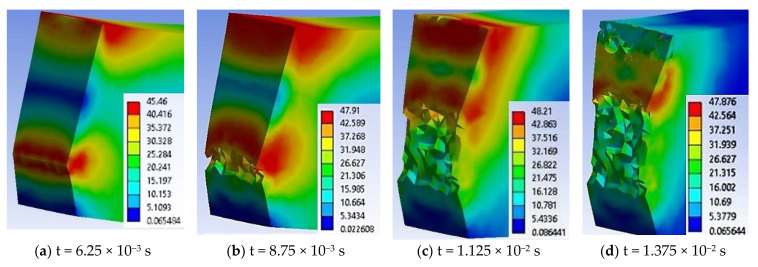
Moments during impact for material PA6m.

**Figure 25 materials-13-05837-f025:**
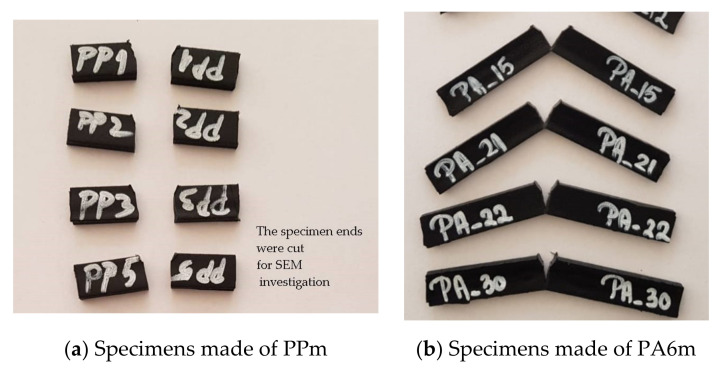
Breaking of specimens following Charpy tests.

**Figure 26 materials-13-05837-f026:**
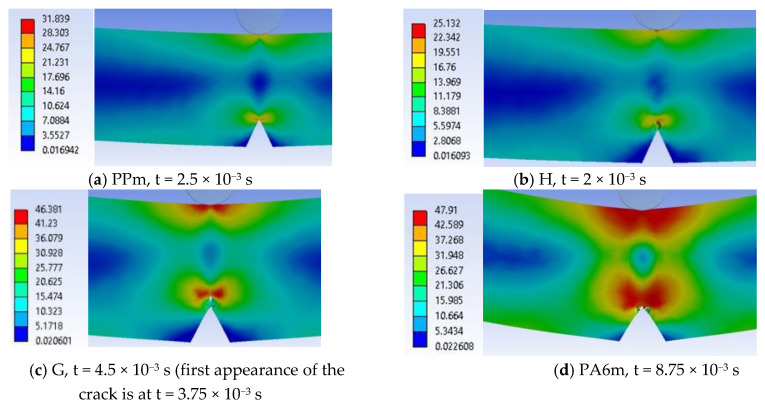
Aspects of crack initiation, for each material and von Mises stress distribution, in MPa.

**Figure 27 materials-13-05837-f027:**
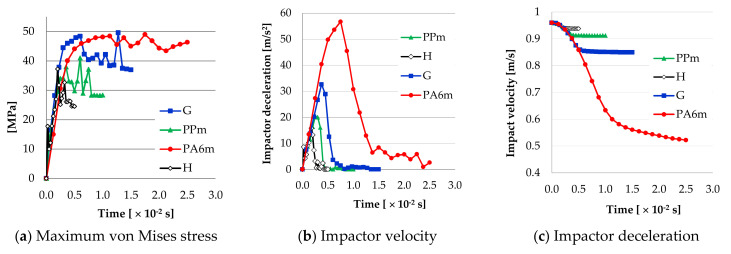
Evolution of several characteristics in time, for each material.

**Figure 28 materials-13-05837-f028:**
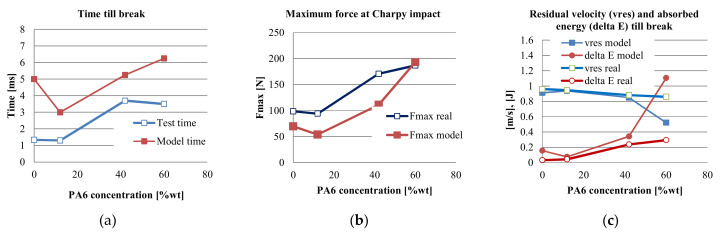
Comparison between simulation and experimental data: (**a**) comparison between test time as time till break in actual tests and the time till break in the simulation; (**b**) comparison between maximum values of recorded force during the test and the force calculated for the simulation; (**c**) comparison between values obtained by tests and those from simulation, for absorbed energy (delta E real and delta E model, respectively) and residual velocity (vres real and vres model, respectively).

**Table 1 materials-13-05837-t001:** Recipes for the formulated polymer blends. Polyamide 6 (PA6), polypropylene (PP), and ethylene propylene diene monomer rubber (EPDM).

Material	PA6	PP	EPDM	Polybond 3200	Kritilen PP940
PPm	-	99	-	-	1
H	12	60	8	20	-
G	42	20	28	10	-
PA6m	60	-	40	-	-

**Table 2 materials-13-05837-t002:** Bodies involved in the model and their nodes and elements.

Body	Numbers of Nodes	Numbers of Elements
Impactor	4259	20,738
Support 1	3996	3146
Support 2	4212	3354
Sample	20,811	106,591
Total model	33,278	133,829

**Table 3 materials-13-05837-t003:** Characteristics of modeled materials.

Characteristic	PPm	H	G	PA6m
Density, kg/m^3^	915	915	915	915
Young modulus, E, MPa	1466	1582	1625	1595
Poisson’s ratio, ν	0.4	0.4	0.4	0.4
Bulk modulus, MPa	2443.3	2636.7	2708.3	2658.3
Shear modulus, MPa	523.57	565	580.36	569.6
EPS at break	0.09	0.036	0.156	0.308

**Table 4 materials-13-05837-t004:** Characteristics obtained from Charpy tests, for the second family of blends.

Characteristic	PPm	H	G	PA6m	PA6
Absorbed energy at break [J]	0.068	0.061	0.277	**0.334**	0.256
Maximum force [N]	98.54	94.05	170.70	**182.52**	200.96
Impact resistance [kJ/m^2^]					
average	2.26	2.017	9.23	**11.47**	8.52
range	1.7–2.8	1.5–2.73	5.76–12.73	**10.16–12.96**	7.0–9.3

**Table 5 materials-13-05837-t005:** Analysis of important moments during impact.

Material	Moment of Impact	Moment of Initiation of the First Crack [s]	Moment of Initiation of the Second Crack [s]	Moment of Total Detachment of Fragments [s]	Total Running Time [s]
PPm	0	2.5 × 10^−3^	3.5–4.0 × 10^−3^	7.0 × 10^−3^	1.0 × 10^−2^
H	0	1.75 × 10^−3^	2.0 × 10^−3^	4.0 × 10^−3^	5.0 × 10^−3^
G	0	3.75 × 10^−3^	6.0 × 10^−3^	9–9.75 × 10^−3^	1.5 × 10^−2^
PA6m	0	7.5 × 10^−3^	1.125 × 10^−2^	not fully detach	2.5 × 10^−2^

**Table 6 materials-13-05837-t006:** Impact characteristics.

The Parameter	PPm	H	G	PA6m
Equivalent plastic strain (EPS) at break [-]	0.09	0.036	0.156	0.308
Maximum von Mises stress during impact [MPa]	41.1	37.2	48.4	48.9

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
