# Peer review of "Numerical and Experimental Results on Charpy Tests for Blends Polypropylene + Polyamide + Ethylene Propylene Diene Monomer (PP + PA + EPDM)"

_materials, 2020, doi:10.3390/ma13245837_

Round 1

Reviewer 1 Report

  • 23 not "was", but "were"
  • 49 correct references 22 and 23 in a single brackets couple and so for lines: 53, 89, 103, 104, 121, 126, 155, 167,
  • 64-73 it would be advisable to reformulate the sentence, removing some brackets to give better text comprhension
  • 84 correct subscript -1
  • 98 replaced point with comma
  • 100-101 remove brackets, it's better a list
  • 104 better not to start a sentence with "but"
  • 126 remove brackets
  • 138 fig.1 should be explained in more detail
  • 146 please, replace "an" with "and"
  • 159 why reference is not numbered?
  • 168 delete "they"
  • 172 delete
  • 175 repetition of "and sample"
  • 188-191 reformulate the sentence, possibly without brackets 
  • 192 remove brackets 
  • 196-197 remove brackets 
  • 204-205 remove brackets and the content
  • 206 a bracket for fig. 5b is missing
  • 206-221 please, reformulate the sentence, it's too long and complex for easy reading
  • 229 where is fig. 7b?
  • 206-240 I think it's better to interrupt the text with figures, it would be easier to follow the results. Please, explain figs. 7 and 8 in more details
  • 281 do you have roughness data?
  • 299-300 reformulate sentences, please, more linear 
  • 307 correct subscript text

NOTES:

  1. Too much text in round brackets makes reading less clear. I recommend rephrasing some sentences and removing the brackets, inserting the content in the text.
  2. The data separated by 3 dots are not very clera, e.g. line 136
  3. Text in round brackets in fig. 21a should be inserted in the figure caption and so in fig. 26c
  4. Referencing figures too far foward or too far back makes reading more difficult. If possible, simplify the figures references in the text, even deleting some figures.

Author Response

I completely revised the paper taking into account your comments.

Dear Reviewer,

All the corrections you had noticed I take them into account and also I noticed several others.

- We modify the text to be more clear and with no sentences too long,

- We remove many brackets, including those you have already noticed,

- We reorganized the figure with too many graphics and I put each graphic near the text,

The data separated by 3 dots are not very clear, e.g. line 136. Now there is “(approx. 35x10-3 s-1 to 150x10-3 s-1),”

Text in round brackets in fig. 21a should be inserted in the figure caption and so in fig. 26c. It was inserted

The referencing figures are now near the text.

Initial lines 490-507 were repeated at 523-540. Now, they are properly introduced.

Please, see the re-formulated manuscript.

The authors

Reviewer 2 Report

This MS reports a numerical and experimental investigation on Charpy tests for blends polypropylene + polyamide + ethylene propylene diene monomer. I think there are many problems in the MS. The MS should be in major revision before considering publication. The comments are as follows:

  • There are a lot of errors in the article, including the EDPM in the title, which should be EPDM. The form of speed unit is not unified. The description chart in the article does not conform to the actual chart, such as Fig. 6 in 222 lines and Fig7b in 229 lines, and EDX analysis in line245. Where is the EDX data??? The writing form of decimal point in Table 2 is not uniform. What sample is G1 in Fig. 25???
  • Polybond 3200 is used in Table 1 and Polybond 3000 is written in line 129. Are these two things the same??
  • From Fig. 9, how can we see that the fracture of PA6m belongs to the fracture of the product in the PA6m lateral view? The author needs to provide the profile of the cross section!
  • Is the unit of impact resistance (kJ/m2) in Table 5 correct? J/m2 is used in other places.
  • Line511-512 the longest time recorded on the simulation for PA6m is t = 2.5x10-2 however, in Fig.22 (d), the value is t = 8.75x10-3 S. which one is correct??
  • In Fig. 6, it is necessary to provide the section morphology images of PP, PPm, H, G, PA and PA6m at the same magnification, which can better interpret the micro morphology and explain that the impact strength of the blends is higher!
  • Line 299-300, the author concluded that “for materials G and PA6m, the results are very good, in the sense that PA6m exceeds PA6, and G also exceeds the energy at break on impact of polyamide”. The author needs to give a reasonable explanation, especially in G, is the role of PP or EPDM? Why?
  • The structure of the article needs to be adjusted. The figures explanation in the conclusion should be put into the discussion part. In addition, the conclusion should be simplified. It is not appropriate to refer to other people's literature in the conclusion, so it should be placed in the data discussion section.
  • The author needs to clarify the ideas and improve English writing. The English writing in the MS is very poor. Many sentences are difficult to understand. There are 26 pictures in this paper, which are too many and can be combined properly or put in the supporting information.

Author Response

This MS reports a numerical and experimental investigation on Charpy tests for blends polypropylene + polyamide + ethylene propylene diene monomer. I think there are many problems in the MS. The MS should be in major revision before considering publication. The comments are as follows:

  • There are a lot of errors in the article, including the EDPM in the title, which should be EPDM. The form of speed unit is not unified. The description chart in the article does not conform to the actual chart, such as Fig. 6 in 222 lines and Fig7b in 229 lines, and EDX analysis in line245. Where is the EDX data??? The writing form of decimal point in Table 2 is not uniform. What sample is G1 in Fig. 25???

Done

  • Polybond 3200 is used in Table 1 and Polybond 3000 is written in line 129. Are these two things the same??

The first is the used compatibilizing agent

  • From Fig. 9, how can we see that the fracture of PA6m belongs to the fracture of the product in the PA6m lateral view? The author needs to provide the profile of the cross section!

As you see in the macro photo in Fig. 21, the specimens made of PA6m are not completely broken

  • Is the unit of impact resistance (kJ/m2) in Table 5 correct? J/m2is used in other places.

Yes, it is.

  • Line511-512 the longest time recorded on the simulation for PA6m is t = 2.5x10-2however, in Fig.22 (d), the value is t = 8.75x10-3 which one is correct??
  • In Fig. 6, it is necessary to provide the section morphology images of PP, PPm, H, G, PA and PA6m at the same magnification, which can better interpret the micro morphology and explain that the impact strength of the blends is higher!

“The SEM images (Figure 10) shows that the introduction of PA6 in higher concentration changed the morphology of the blend. Material H (Figure 10a) has a PP matrix with PA6 droplets and pores of the order of a few microns. The matrix has a brittle fracture. The droplets located on the breaking zone are very deformed, elongated and the matrix breaks as a more brittle polymer, with linear micro-flows, probably as a result of a lower crystallinity of PP. Material G (Figure 10b) has the inverted phases, as it resulted from the EDX analysis. The PP droplets are larger and do not have cohesion on the whole surface with the matrix.

a) Material H

b) Material G

c) Material PA6m

d) Material PA6m (detail)

Figure 10. Fracture morphology, obtained for Charpy test.

Analyzing both SEM images for material PA6m (Figure 10 c and d), this blend of PA6 and EPDM seems to be a homogeneous one: a) a low magnification of the arrested crack and b) a detail that point out that the blend is homogenous and extensively deformed.”

  • Line 299-300, the author concluded that “for materials G and PA6m, the results are very good, in the sense that PA6m exceeds PA6, and G also exceeds the energy at break on impact of polyamide”. The author needs to give a reasonable explanation, especially in G, is the role of PP or EPDM? Why?

For material G, EPDM and PA6 formed a matrix more ductile that that in material H and the droplets of PP are better fixed in this matrix. The material PA6m being a blend of PA6+EPDM, has homogenous aspect and the detail in Figure 10d evidences extensive deformation, but not completely break.

  • The structure of the article needs to be adjusted. The figures explanation in the conclusion should be put into the discussion part. In addition, the conclusion should be simplified. It is not appropriate to refer to other people's literature in the conclusion, so it should be placed in the data discussion section.

Done.

  • The author needs to clarify the ideas and improve English writing. The English writing in the MS is very poor. Many sentences are difficult to understand. There are 26 pictures in this paper, which are too many and can be combined properly or put in the supporting information.

We revised the English writing and the figures are now put near the text.

Thank you for your comments.

Please, see the re-formulated manuscript.

Round 2

Reviewer 2 Report

After the revision, the level of the MS has been significantly improved, but there are still several questions to be clarified.

  • Line 250-251, the author concluded that the phases inverted from the EDX analysis, where is your EDX data? Please add the data.
  • Line 302, Pa6m should be written as PA6m.
  • In Figure 27, the lines of the four materials (G, PPm, PA6m, H) should be used the same curve color and sample.

Author Response

Dear Madame/ Dear Sir,

I attach a document with EDX analysis. As you see, the quality of images is not so good (especially for material H) as the other SEM images in the paper and I will try to do the analysis again.

Thank you for your suggestions for graphs.
